# Reasoning Matters: Benchmarking and Advancing Spatial Reasoning in Vision-Language Models via Agentic Approaches

## Abstract

CAPTCHA, originally designed to distinguish humans from robots, has evolved into a real-world benchmark for assessing the spatial reasoning capabilities of vision-language models. In this work, we first show that step-by-step reasoning is crucial for vision-language models (VLMs) to solve CAPTCHAs, which represent high-difficulty spatial reasoning tasks, and that current commercial vision-language models still struggle with such reasoning. In particular, we observe that most commercial VLMs (e.g., Gemini, Claude, GPT, etc.) fail to effectively solve CAPTCHA and thus achieve low accuracy($\sim$ **21.9%**), but our findings indicate that requiring the model to perform step-by-step reasoning before generating the final coordinates can significantly enhance its solving accuracy, this underscoring the severity of the gap. To systematically study this issue, we introduce **CAPTCHA-X**, the first real-world CAPTCHA benchmark with reasoning, covering seven categories of CAPTCHAs (e.g., Gobang, Hcaptcha, etc) with step-by-step action solutions, and grounding annotations. We further define five reasoning-oriented metrics that enable a comprehensive evaluation of models' reasoning capabilities. To further verify the effectiveness of reasoning, we propose a general agentic VLMs-based framework, incorporating the reasoning abilities of the model itself. Our method achieves state-of-the-art performance across five high-difficulty CAPTCHA types in general agents, with an average solving accuracy of **83.9%**, substantially surpassing existing baselines. These results both reveal the limitations of current models and highlight the importance of reasoning in advancing visual-spatial challenges in the future.

## 1 Introduction

CAPTCHAs were originally introduced as a security mechanism to distinguish humans from machines (Von Ahn et al., 2003). Early text-based CAPTCHAs exploited the limits of OCR (Wang et al., 2018b), but advances in computer vision shifted them toward complex visual–spatial puzzles requiring spatial reasoning, 3D mental rotation, and multi-step inference (Gao et al., 2021a; Luo et al., 2025). This evolution transforms CAPTCHAs from perception tests into probes of higher-level cognition, serving both as defenses against automated attacks and as testbeds for machine reasoning (Ding et al., 2025). Today, they stand as real-world benchmarks for evaluating spatial intelligence in vision–language models, combining perception, reasoning, and decision-making (Liu et al., 2023).

With the rapid progress of vision–language models (VLMs), existing CAPTCHA benchmarks suffer from several fundamental limitations. While Open CaptchaWorld (Luo et al., 2025) introduces reasoning-related difficulty metrics, it lacks reasoning annotations, preventing a comprehensive evaluation of models' reasoning abilities. Meanwhile, many recent general solvers (e.g., Halligan) achieve strong performance by combining VLMs with auxiliary tools and finetuned model (Teoh et al., 2025) (Deng et al., 2024) (Wu et al., 2025), yet they do not explicitly incorporate reasoning, and the lack of reasoning annotations further obscures the intrinsic reasoning capacity of the underlying models. Besides, most other datasets only provide CAPTCHA images with corresponding answers (such as coordinates) and evaluate correctness by measuring whether the distance between predicted and ground truth values falls within an empirically set threshold. This mismatch often

yields offline results that fail to reflect online performance and fail to capture the reasoning processes underlying successful CAPTCHA solving, as we will discuss in detail in §3.1. Ultimately, a central gap remains: no prior work has definitively answered whether reasoning itself is the key to solving CAPTCHA.

In this paper, we create the first real-world benchmark CAPTCHA-X with reasoning and show evidence that reasoning is the key to solving CAPTCHAs. Directly applying commercial VLMs to solve CAPTCHAs, especially highly difficult tasks, achieves only an accuracy of 21.9%. underscoring severe deficits in spatial reasoning. As shown in Figure 1, we have seven categories CAPTCHA collection.

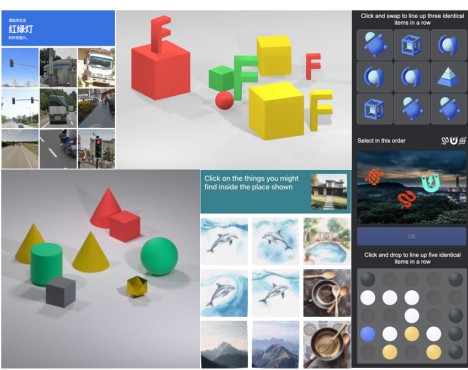

Once reasoning is introduced, however, performance statistically significantly improves by an average of 27.5% relative to the non-reasoning baseline. This confirms that reasoning fundamentally changes models' reasoning accuracy. To further validate this finding, we design an

Figure 1: Our CAPTCHA-X Benchmark.

agentic VLM approach that relies only on large models with reasoning, without requiring complex toolchains or task-specific fine-tuned models.

Our contributions can be summarized as follows:

- We introduced CAPTCHA-X, the first real-world CAPTCHA benchmark with reasoning. CAPTCHA-X covers seven challenges with authentic annotations, region-level acceptance zones, and reasoning steps to systematic evaluation of reasoning capability for VLMs.
- Using CAPTCHA-X, we demonstrated the importance of reasoning for CAPTCHA solving and exposed severe deficits in existing VLMs' spatial reasoning capability.
- Experiments on our benchmark show that incorporating reasoning improves performance by 27.5% relative to the baseline, and statistical analysis confirms the improvement is highly significant ($p < 0.001$), providing the first systematic evidence that reasoning fundamentally improves model accuracy.
- To further validate our finding, we propose a general agentic VLM framework that operationalizes the model's reasoning process through a structured pipeline, enabling robust CAPTCHA solving without auxiliary components or task-specific adaptations. This framework serves as a conceptual validation that reasoning alone suffices to solve real-world CAPTCHAs. On our CAPTCHA-X, this design achieves an average accuracy of 83.9% across seven CAPTCHA categories and sets new state-of-the-art results on five categories in general solving agents.

## 2 RELATED WORK

**CAPTCHA Evolution and Benchmarking.** Over two decades, CAPTCHAs evolved from distorted text (Von Ahn et al., 2003) to image-based challenges like Asirra, later broken by machine learning (Hitaj et al., 2020). This fragility spurred variants requiring logical reasoning and multi-step interaction. Recent benchmarks such as MCA-Bench (Wu et al., 2025) and Bot-Hard (Teoh et al., 2025) emphasize multimodal reasoning and robustness, framing CAPTCHAs as tests of spatial intelligence. Yet, as Table 1 shows, gaps remain: Open CaptchaWorld (Luo et al., 2025) uses synthetic data without reasoning labels; Halligan (Teoh et al., 2025) and OEDIPUS (Deng et al., 2024) provide real data but lack reasoning annotations; and MCA-Bench, though large, is synthetic and detached from real-world challenges. By contrast, our CAPTCHA-X is one of the few large-scale real-world datasets (1,839 puzzles), and uniquely enriched with detailed reasoning annotations and region-based validation. This makes it the first benchmark to evaluate both solving accuracy and reasoning in vision–language models under realistic conditions.

**Reasoning in Visual CAPTCHA Solving.** Reasoning has become a decisive factor in solving modern CAPTCHAs. Early VLM-based solvers emphasized perceptual accuracy but failed on tasks requiring spatial inference or multi-step logic (Shi et al., 2019). Later work explored adversarial and

Table 1: CAPTCHA Benchmark Comparisons.

| Benchmark | Real world | Reasoning | Region Consistent | Scale |
|---|---|---|---|---|
| Open CaptchaWorld (Luo et al., 2025) | ✗ | ✗ | ✗ | 225 |
| Halligan (Teoh et al., 2025) | ✓ | ✗ | ✓ | 2600 |
| OEDIPUS (Deng et al., 2024) | ✓ | ✗ | ✗ | 300 |
| MCA-Bench (Wu et al., 2025) | ✗ | ✗ | ✗ | 180000 |
| CAPTCHA-X (Ours) | ✓ | ✓ | ✓ | 1839 |

cognitive-inspired CAPTCHA designs, showing that robustness depends not only on recognition but also on following reasoning chains (Bursztein et al., 2011; Yan & El Ahmad, 2016). Recent methods employ large language models to guide multi-modal perception, yet their evaluation usually reports only final accuracy without reasoning annotations or ablations (Ye et al., 2022). Platforms like Open CaptchaWorld attempted to capture reasoning complexity with new metrics and task designs, but still lacked reasoning annotations, limiting comprehensive evaluation across models.

**Spatial Reasoning Benchmarks.** Spatial reasoning is central to visual intelligence, motivating benchmarks such as ARC-AGI (Chollet, 2019) with grid-based puzzles testing object permanence and spatial relations, CLEVR (Johnson et al., 2017) for compositional reasoning, and PTR (Hong et al., 2021) for part-whole hierarchies. Extending to 3D, 3DSRBench (Ma et al., 2024) exposes large human–machine gaps. Distinctly, our CAPTCHA benchmark leverages decades of adversarially tested human–machine challenges, offering spatial reasoning tasks inherently designed to reveal AI weaknesses.

## 3 METHOD

### 3.1 DATA COLLECTION AND CURATION

To address the limitations of existing benchmarks, we developed CAPTCHA-X through a systematic data collection pipeline with high-quality, reasoning steps annotations.

**Data Collection.** We collect CAPTCHA data by programmatically interacting with websites using Selenium (Jason Huggins) and PyAutoGUI (Sweigart), while recording comprehensive mouse action sequences and screenshots before and after each puzzle. The detailed data collection process is provided in §A.1.

**Grounding Annotation Generation.** After solving a CAPTCHA, we record the click coordinates, which may not fall exactly at the object center. We therefore define acceptance regions by manually marking all valid circles or boxes and count a click as correct if it falls within one of them. Unlike prior work that uses a fixed threshold around the click, our approach covers the full target area more reliably, as shown in Figure 2.

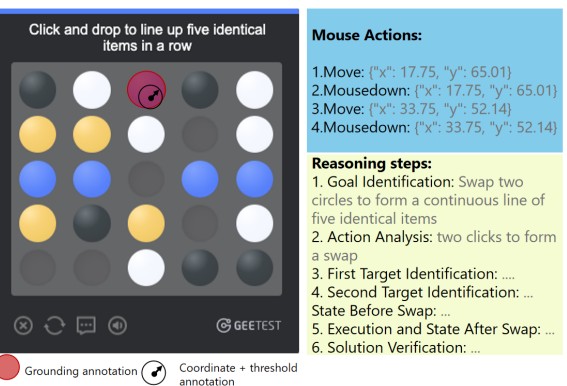

Figure 2: Grounding annotation (red) versus threshold-based annotation (black) in a **GeeTest Gobang** puzzle, along with recorded mouse actions and reasoning steps. These mouse actions and reasoning steps are generated by using carefully designed prompts.

**Reasoning Steps Generation.** To create reasoning annotations with accurate mouse actions, we use LLMs (i.e., GPT-5) to generate step-by-step reasoning steps. We choose LLM-based generation because manual annotation is highly labor-intensive, and manually written reasoning steps tend to lack diversity. Concretely, we condition the LLM on the ground-truth action trajectory for each puzzle and employ carefully designed prompts that are (1) goal-directed, explicitly stating the CAPTCHA's objective and required click targets, (2) vision-language aware, maximally exploiting the LLM's ability to jointly process visual content and text, (3) naturally expressed, encouraging concise and conversational reasoning steps, and (4) challenging, designed to maximally elicit the model's reasoning ability. The prompt template is provided in §A.3.

**Quality Assurance.** To ensure the reliability and accuracy of CAPTCHA-X, every generated reasoning step underwent rigorous human verification by four domain experts. Each expert independently scored the quality of the reasoning steps on a 0–10 scale. If the score difference among the experts exceeded 2 points, or if the average score fell below 5, the sample was jointly re-examined. Expert agreement reached 98% under this criterion, and the remaining cases were resolved through discussion, yielding 100% consensus in the final annotations. This multi-expert verification process ensures that CAPTCHA-X provides a robust and trustworthy foundation for evaluating the spatial reasoning capabilities of vision-language models.

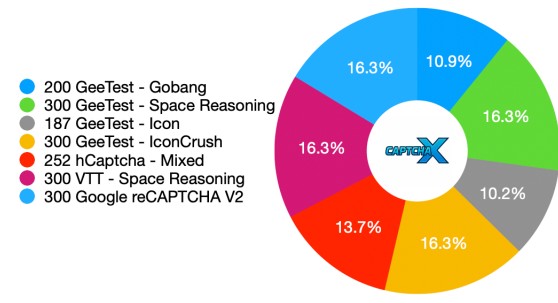

- ● 200 GeeTest - Gobang
- ● 300 GeeTest - Space Reasoning
- ● 187 GeeTest - Icon
- ● 300 GeeTest - IconCrush
- ● 252 hCaptcha - Mixed
- ● 300 VTT - Space Reasoning
- ● 300 Google reCAPTCHA V2

Figure 3: Distribution of our benchmark.

**CAPTCHA-X.** Our benchmark comprises 1,839 CAPTCHA puzzles across seven categories, as shown in Figure 3. It covers grid-based puzzles, spatial reasoning tasks, and mixed styles, with each category contributing about 10–16% of the total for balanced distribution. For every puzzle, we provide reasoning steps and mouse action sequences to evaluate both solving accuracy and reasoning quality. An example from Gobang is shown in Figure 2.

## 3.2 CAPTCHA EVALUATION METRICS

To systematically evaluate models' capability in solving CAPTCHAs, we define a comprehensive evaluation metric. Specifically, our metrics consider both the correctness of actions and the reasoning by comparing with our annotated ground truth.

We formalize the answer to a CAPTCHA puzzle as an ordered sequence:

$$\mathcal{S} = \{(a_1, c_1), (a_2, c_2), \ldots, (a_m, c_m); R\}, \tag{1}$$

where $(a_i, c_i)$ denotes the $i$-th action and its associated coordinate; $R = \langle r_1, r_2, \ldots, r_k \rangle$ denotes the reasoning process, expressed as a sequence of steps.

### 3.2.1 ACTION ACCURACY

Our metric measures if the predicted action–coordinate sequence $\{(a_1, c_1), (a_2, c_2), \ldots, (a_N, c_N)\}$ exactly matches the ground-truth sequence in both order and correctness. Let $a_i^*$ denote the ground-truth action at step $i$, $(\hat{x}_i, \hat{y}_i)$ denote the predicted coordinate $c_i$, and $\mathcal{RG}_i$ the corresponding acceptance region. We define sequence-level accuracy as:

$$AccRate = \frac{1}{M} \sum_{j=1}^{M} \mathbf{1}\left(a_i^{(j)} = a_i^{*(j)} \ \wedge \ (\hat{x}_i^{(j)}, \hat{y}_i^{(j)}) \in \mathcal{RG}_i^{(j)}, \ \forall i\right), \tag{2}$$

where $M$ is the total number of CAPTCHA puzzles. Here $\mathbf{1}\{\cdot\}$ returns 1 only if the entire predicted sequence exactly matches the ground truth in both action order and coordinates, and 0 otherwise.

### 3.2.2 REASONING ACCURACY

To comprehensively evaluate the quality of model-predicted reasoning, we design multiple new metrics for reasoning, each motivated by a distinct aspect of reasoning quality. We argue that high-quality reasoning steps should achieve high solving accuracy or capture maximal complexity with minimal reasoning cost.

**Reasoning Steps.** To measure the granularity of reasoning, we count the number of reasoning steps in the generated textual reasoning. Since our reasoning is expressed as step-by-step text, this metric naturally reflects the level of detail in the reasoning process. A larger number of steps typically implies a more complex reasoning trajectory, but also indicates reduced reasoning efficiency.

**Reasoning Length.** We measure the total number of tokens in the generated reasoning text. In contrast to Reasoning Steps, which capture the structural depth of reasoning, this metric quantifies the overall textual length, offering a finer-grained view of reasoning cost.

**Reasoning Score.** To evaluate alignment with ground-truth reasoning, we use four different large language models (LLMs) to provide automatic scores. Following the HD-Eval framework (Li et al., 2023), the evaluation is decomposed into multiple sub-dimensions to reduce potential bias. Formally, if $s_{i,m}$ denotes the score for instance $i$ from model $m$, then

$$S_i = \frac{1}{M} \sum_{m=1}^{M} s_{i,m}, \quad M = 4. \tag{3}$$

To verify that LLM-based evaluation is consistent with human judgment, we randomly sampled 5% of the instances from each CAPTCHA category and asked human experts to provide independent scores. The Pearson correlation between the aggregated LLM scores and human scores reached **0.92**, indicating that our automatic evaluation method is well aligned with human preference.

**Reasoning Efficiency.** To assess the trade-off between predictive accuracy and reasoning cost, we define an efficiency metric. Let $Acc_i$ denote the accuracy of model $i$, $\hat{L}_i = L_i/\overline{L}$ the normalized reasoning length, and $\hat{S}_i = S_i/\overline{S}$ the normalized reasoning steps. With equal weights $\alpha = \beta = 0.5$, efficiency is computed as

$$Efficiency_i = \frac{Acc_i}{\alpha \cdot \hat{L}_i + \beta \cdot \hat{S}_i}. \tag{4}$$

Values are further using min–max normalized to $(0, 1)$. In all, higher reasoning efficiency reflects the model achieving stronger accuracy with fewer steps or tokens, which is more efficient.

**Trajectory Complexity Index (TCI).** To quantify the structural complexity of reasoning trajectories, we capture linguistic signals such as backtracking words (*but*, *however*, etc.) and symbolic markers (coordinates, grid indices, etc.). For each instance $j$ in group $i$, we aggregate feature counts $F_{i,j}$ and normalize them by group-level averages:

$$z_{i,j} = \frac{\sum_F (F_{i,j} - \overline{F}_i)}{0.5 \cdot (s_i/\overline{s}) + 0.5 \cdot (t_i/\overline{t})}. \tag{5}$$

The final TCI is obtained by applying a sigmoid function, which maps the feature values into the normalized range of (0, 1):

$$TCI_i = \sigma\left(\frac{1}{N_i} \sum_{j=1}^{N_i} z_{i,j}\right), \quad \sigma(x) = \frac{1}{1 + e^{-x}}. \tag{6}$$

A higher TCI indicates frequent backtracking or symbolic reasoning, demonstrating the intrinsic complexity of the reasoning path, and also reflecting higher information density.

### 3.3 VISION-LANGUAGE MODEL AGENTIC PIPELINE

To further validate our findings, we introduce a novel agentic framework that, unlike prior solvers, relies solely on a VLM's inherent reasoning ability without external toolchains or fine-tuned models as shown in Figure 4.

The pipeline begins with a **Category Judger** that routes each puzzle to either a grid-based or a non-grid-based branch. This classification is crucial because the two types of puzzles require fundamentally different reasoning strategies. And all the clickable CAPTCHA can be divided into these two categories. For grid-based puzzles (e.g., Google reCAPTCHA, GeeTest IconCrush), a dedicated **Mapping Tool**, implemented as a large language model guided by carefully designed prompts, converts the puzzle board into an $A \times A$ symbolic grid (e.g., $[a, a, a; b, b, c; c, b, b]$). This abstraction enables the **Reasoning Steps Generator** to conduct structured step-by-step inference over the grid, leading to accurate identification of the target cell(s). In contrast, non-grid-based puzzles (e.g., GeeTest Icon, VTT Space Reasoning) rely on spatial semantics rather than grid indexing, and therefore the **Reasoning Steps Generator** first produces reasoning steps that are refined by a **Spatial Understanding Expert**, which grounds objects and regions into spatial coordinates. To ensure logical consistency across both branches, a **Discriminator** validates that the generated reasoning is coherent before passing it forward. The validated reasoning is then handled by an **Action Generator**, which translates reasoning outputs into executable click coordinates. Finally, an **Action Executor** performs the actual clicks on the screen to solve the CAPTCHA. By explicitly distinguishing between grid-based and non-grid-based categories, this unified framework highlights the central role of reasoning in solving diverse visual CAPTCHA.

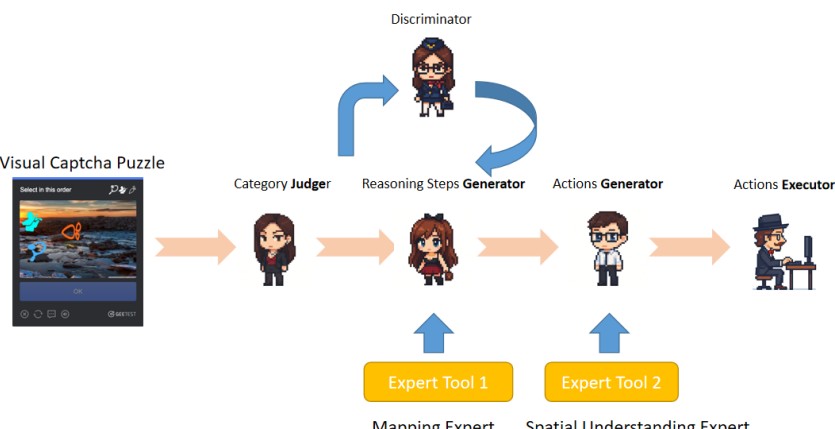

Figure 4: Our Agentic Vision-Language Model Pipeline.

## 4 EXPERIMENTS

We conduct experiments to assess the role of reasoning in CAPTCHA solving by comparing model performance with and without reasoning and measuring spatial alignment via $L_2$ distance. All experiments use a fixed API configuration (temperature $= 0$, seed $= 41$) for reproducibility. We report results in two dimensions: **Action Evaluation**, which measures end-task accuracy, and **Reasoning Evaluation**, which analyzes the quality of intermediate reasoning steps.

### 4.1 ACTION EVALUATION

**Evaluation of Prediction Accuracy.** As shown in Table 2, prompting models to generate reasoning steps almost always improves **solving accuracy**, confirming that reasoning provides strong guidance for CAPTCHA solving. Figure 5 further illustrates this trend.

Table 2: Model performance (WR = With Reasoning, WOR = Without Reasoning) across different CAPTCHA types.

| Model | Gobang | | Icon | | Iconcrush | | Recaptcha | | Space Reasoning | | hcaptcha | | VTT | |
|---|---|---|---|---|---|---|---|---|---|---|---|---|---|---|
| | WR | WOR | WR | WOR | WR | WOR | WR | WOR | WR | WOR | WR | WOR | WR | WOR |
| *GPT Family* | | | | | | | | | | | | | | |
| GPT-O3 | 2.00 | 0.00 | 22.00 | 29.79 | 3.67 | 3.67 | 10.67 | 1.82 | 10.00 | 1.50 | 27.67 | 0.00 | 7.00 | 3.67 |
| GPT-4O | 0.00 | 0.00 | 9.52 | 7.48 | 28.00 | 23.33 | 11.00 | 1.52 | 47.00 | 40.00 | 23.71 | 1.92 | 42.00 | 37.67 |
| GPT-5-Nano | 0.00 | 0.00 | 0.00 | 0.00 | 28.00 | 23.33 | 8.33 | 2.00 | 31.00 | 32.00 | 58.33 | 40.00 | 30.67 | 32.67 |
| *Gemini Family* | | | | | | | | | | | | | | |
| Gemini-2.5-Pro | 57.00 | 48.00 | 59.30 | 46.30 | 75.00 | 66.67 | 64.00 | 56.52 | 68.00 | 64.67 | 80.95 | 81.35 | 63.00 | 56.00 |
| Gemini-2.0-Flash | 2.00 | 0.00 | 36.33 | 39.67 | 2.33 | 2.00 | 36.33 | 31.67 | 53.00 | 51.00 | 43.21 | 0.79 | 45.67 | 47.67 |
| *Other Models* | | | | | | | | | | | | | | |
| Claude-4-Opus | 18.00 | 8.00 | 17.65 | 13.00 | 18.00 | 6.67 | 12.33 | 3.33 | 29.00 | 23.33 | 26.70 | 0.00 | 26.67 | 23.67 |
| Qwen-2.5VL-72B | 0.00 | 0.00 | 0.00 | 0.00 | 6.00 | 5.00 | 14.00 | 0.00 | 24.00 | 27.67 | 38.10 | 36.11 | 19.33 | 26.67 |
| *Ours* | | | | | | | | | | | | | | |
| Captcha-X-Agent-O3 (Ours) | 39.00 | – | **80.10** | – | **93.00** | – | 69.40 | – | 96.67 | – | 91.74 | – | 79.00 | – |
| Captcha-X-Agent-2.5-Pro (Ours) | **67.44** | – | 78.60 | – | 92.33 | – | **73.00** | – | **98.67** | – | **94.44** | – | **80.67** | – |

Gemini-2.5-Pro achieves the highest accuracy among existing models, with Gemini-2.0-Flash and GPT-5-Nano following at moderate levels. Claude-4-Opus, GPT-4O, GPT-O3, and Qwen-2.5VL-72B also benefit from reasoning, though with lower absolute performance. Building on GPT-O3 and Gemini-2.5-Pro, our agentic pipeline achieves the best accuracy across all CAPTCHA categories.

**Evaluation of L2 Distance.** Beyond accuracy, our dataset provides region centers to compute $L_2$ distance between predictions and ground truth. This metric directly measures spatial grounding: smaller distances indicate precise localization, while high accuracy with large distances may reflect

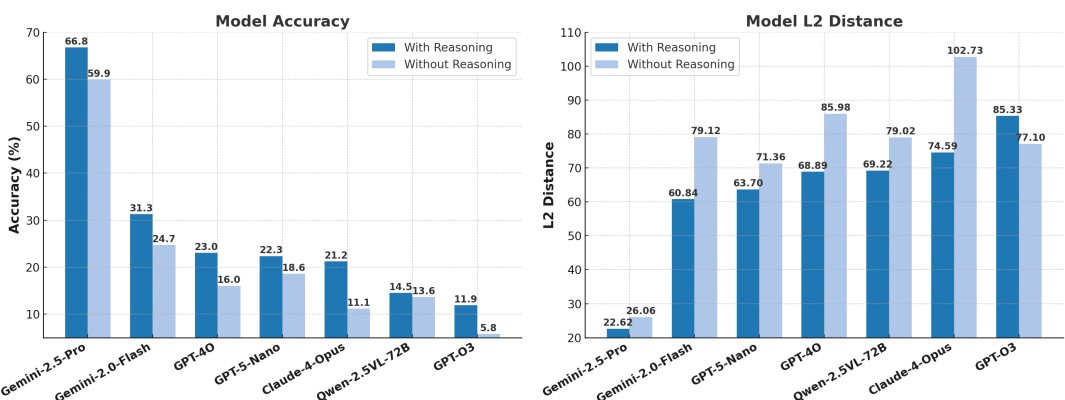

Figure 5: Model Accuracy and L2 Distance with and without reasoning.

Table 3: L2 distance between predicted coordinates and ground-truth centers across CAPTCHA benchmarks (lower is better).

| Model | Gobang | | Icon | | Iconcrush | | Recaptcha | | Space Reasoning | | hcaptcha | | VTT | |
|---|---|---|---|---|---|---|---|---|---|---|---|---|---|---|
| | WR | WOR | WR | WOR | WR | WOR | WR | WOR | WR | WOR | WR | WOR | WR | WOR |
| *GPT Family* | | | | | | | | | | | | | | |
| GPT-O3 | 149.54 | 65.73 | 27.56 | 17.34 | 127.71 | 131.78 | 14.64 | 17.25 | 99.89 | 102.69 | 67.62 | 90.07 | 110.37 | 114.82 |
| GPT-4O | 134.22 | 199.17 | 24.28 | 25.72 | 125.47 | 131.24 | 13.51 | 20.65 | 48.67 | 53.26 | 87.41 | 111.51 | 48.64 | 60.29 |
| GPT-5-Nano | 135.06 | 151.10 | 30.72 | 34.87 | 104.87 | 120.52 | 13.04 | 16.43 | 56.88 | 55.70 | 48.44 | 60.18 | 56.91 | 60.73 |
| *Gemini Family* | | | | | | | | | | | | | | |
| Gemini-2.5-Pro | 19.13 | 27.75 | 8.63 | 9.25 | 34.67 | 38.94 | 3.41 | 2.65 | 34.23 | 34.32 | 18.12 | 27.98 | 40.18 | 41.56 |
| Gemini-2.0-Flash | 120.72 | 148.35 | 12.86 | 18.36 | 134.93 | 128.94 | 9.54 | 14.83 | 40.74 | 41.67 | 57.87 | 153.26 | 49.25 | 48.41 |
| *Other Models* | | | | | | | | | | | | | | |
| Claude-4-Opus | 182.58 | 233.48 | 24.24 | 35.76 | 101.19 | 154.65 | 31.06 | 25.47 | 59.26 | 63.06 | 63.98 | 134.00 | 59.83 | 72.67 |
| Qwen-2.5VL-72B | 121.87 | 129.72 | 29.37 | 30.37 | 126.29 | 163.97 | 13.97 | 21.02 | 62.67 | 68.16 | 58.98 | 62.93 | 71.42 | 76.95 |
| *Ours* | | | | | | | | | | | | | | |
| Captcha-X-Agent-O3 (Ours) | **29.87** | – | 5.19 | – | 26.48 | – | **2.52** | – | **1.15** | – | **8.33** | – | 3.94 | – |
| Captcha-X-Agent-2.5-Pro (Ours) | 37.12 | – | **5.03** | – | **22.32** | – | 2.91 | – | 1.34 | – | 9.74 | – | **3.47** | – |

boundary luck. Using both accuracy and $L_2$ distance yields a more reliable measure of solving quality.

As shown in Table 3, Gemini-2.5-Pro achieves the smallest $L_2$ distances among existing models, with Gemini-2.0-Flash also showing relatively strong spatial grounding. In contrast, weaker models such as GPT-O3 and Claude-4-Opus exhibit very large errors, exceeding 100 pixels in several cases. Notably, our agent consistently achieves the lowest $L_2$ distances across all CAPTCHA types, demonstrating superior localization. These results confirm that $L_2$ distance provides complementary evidence of grounding beyond solving accuracy.

To further validate this relationship, we plot the average performance of all models across all CAPTCHA types in Figure 6. The regression analysis reveals a very strong correlation: models with higher solving accuracy consistently achieve smaller $L_2$ distances.

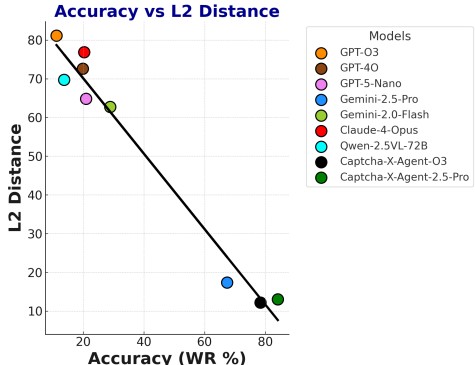

Figure 6: Average Accuracy vs L2 Distance.

Importantly, no outliers are observed, indicating that this pattern holds universally across all tested models.

**Statistical Validation.** For solving accuracy, we adopt McNemar's test (McNemar, 1947), which is designed for paired binary outcomes, and obtain a highly significant result ($p < 0.001$). For $L_2$ distance, we apply the Wilcoxon signed-rank test, and also obtain $p < 0.001$. Moreover, regression analysis between accuracy and $L_2$ distance yields a strong negative correlation with $R^2 = 0.97$ and $p < 0.001$, confirming that higher accuracy is consistently associated with smaller localization

errors. On average, reasoning improves solving accuracy by 27.5% while reducing $L_2$ distance by 14.6%, further validating its effectiveness. Together, these results provide strong statistical evidence that reasoning significantly improves both solving accuracy and spatial localization.

## 4.2 REASONING EVALUATION

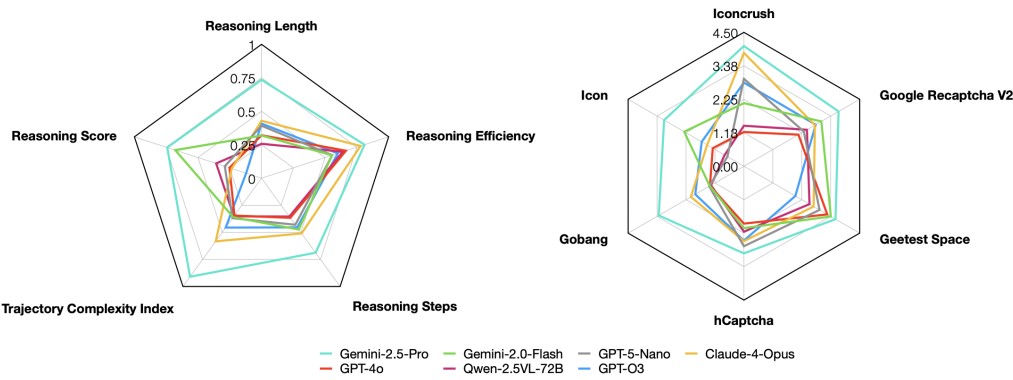

Figure 7: Reasoning Evaluation with Multi-Dimensions: The left radar chart shows overall reasoning metrics averaged across CAPTCHA categories. The right radar chart reports reasoning scores by CAPTCHA type.

To systematically assess reasoning quality, we evaluate multiple reasoning metrics here. Figure 7 presents two complementary radar charts: the left radar chart aggregates overall reasoning metrics averaged across all CAPTCHA categories, while the right radar chart highlights reasoning scores per individual CAPTCHA type. For clarity, we only report the aggregated trends here, while the full quantitative results for all metrics and captcha types are provided in the §A.2.

**Overall Reasoning Metrics.** The left radar chart summarizes average The left radar chart shows average reasoning behaviors across models. Gemini-2.5-Pro is the strongest, combining long and information-dense reasoning with the highest efficiency. Claude-4-Opus ranks second but is much less efficient, while Gemini-2.0-Flash achieves comparable efficiency with shorter reasoning. In contrast, weaker models such as Qwen-2.5VL-72B produce short and low-efficiency traces, indicating limited reasoning capacity.

**Reasoning Score by CAPTCHA Type.** The right radar chart shows reasoning alignment across CAPTCHA types. Gemini-2.5-Pro achieves the highest scores overall, demonstrating strong reasoning quality and generalization. Claude-4-Opus ranks second but with notable drops on some tasks, while GPT-O3 and GPT-4O remain inconsistent. Qwen-2.5VL-72B performs the weakest, rarely exceeding a score of 2.0.

**Correlation Analysis of Reasoning Metrics.**

We conduct a correlation analysis across seven models to verify the validity of our proposed metrics. As shown in Figure 8, Reasoning Score ($r = 0.88$) and Efficiency ($r = 0.82$) both correlate strongly with accuracy, confirming that they are meaningful predictors of task performance rather than ad-hoc measures. Other metrics such as Length, Steps, and TCI

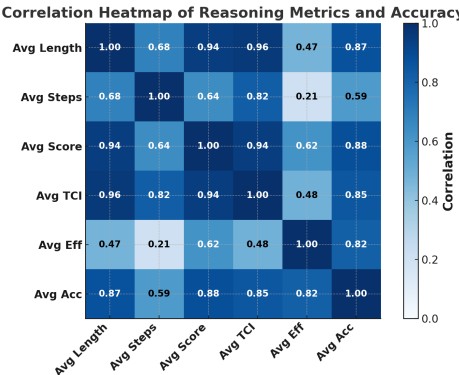

Figure 8: Correlation Heatmap.

capture complementary aspects of reasoning complexity, further supporting the effectiveness of our metric design.

**Reasoning Scaling Law in CAPTCHA.**

Table 4: Comparison of CAPTCHA Solving Accuracy for Different CAPTCHA Solvers

| Model | Icon | Space Reasoning | VTT | Iconcrush | hCaptcha | Gobang | Google Recaptcha V2 |
|---|---|---|---|---|---|---|---|
| *Baseline Models* | | | | | | | |
| Baseline | 46.3 | 64.67 | 50.00 | 66.7 | 0 | 48 | 56.52 |
| OEDIPUS-DSL | – | 65.4 | – | 67.4 | – | 80.2 | – |
| Halligan | 46 | – | 23 | **98** | 82 | **92** | 68 |
| VTTsolver (Gao et al., 2021b) | – | 90.8 | 50 | – | – | – | – |
| PhishDecloaker (Teoh et al., 2024) | – | – | – | – | 74 | – | 72 |
| *Ours* | | | | | | | |
| Captcha-X-Agent (Ours) | **80.1** | **98.67** | **80.67** | 93 | **94.44** | 67.44 | **73** |

Our analysis reveals a linear reasoning scaling law consistently observed across all evaluated models, showing that reasoning score grows proportionally with both reasoning length and trajectory complexity. Specifically, we observe a near-perfect linear fit, e.g., Length $\approx 78.95 \cdot$ Score $- 62.11$ ($p < 0.01$ in significance test) and TCI $\approx 0.349 \cdot$ Score $- 0.333$ ($p < 0.01$), across diverse models. Since reasoning score strongly predicts task accuracy ($r = 0.88$), this law establishes a principled connection between reasoning cost and problem-solving ability, enabling accuracy to be forecasted directly from reasoning complexity (Figure 9).

### 4.3 AGENTIC EVALUATION

We evaluate both a direct-prediction baseline and our proposed reasoning-centric agentic pipeline for CAPTCHA solving. The baseline uses Gemini-2.5-Pro without reasoning, where the model directly outputs click coordinates from the CAPTCHA image.

Among prior solvers on our dataset, Halligan (tool-integrated) and OEDIPUS (fine-tuned) are the only general agent models available for comparison. In contrast, our agent achieves state-of-the-art performance on five out of seven tasks (Table 4), with 98.67 on Space Reasoning, 80.67 on VTT, 94.44 on hCaptcha, 80.1 on Icon, and 73 on Google Recaptcha V2, while also remaining competitive on Iconcrush (93) and Gobang (67.44). These results highlight that our approach achieves strong performance across all CAPTCHA types without toolchains or task-specific finetuning, underscoring reasoning as the key capability for modern CAPTCHA solving.

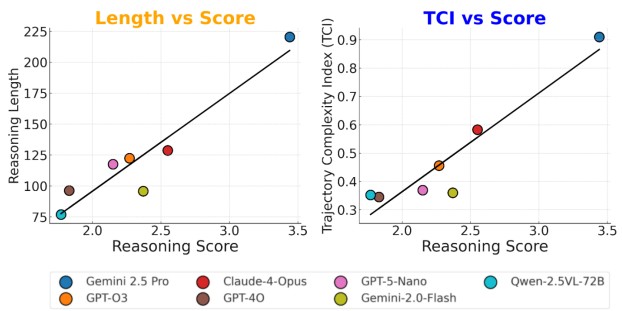

Figure 9: Reasoning Scaling Law in CAPTCHA.

## 5 LIMITATION

While our work highlights the role of reasoning in improving CAPTCHA-solving accuracy, it also raises security concerns. Our results suggest that modern vision–language models can bypass many existing CAPTCHA designs, indicating that CAPTCHAs may soon lose their effectiveness as a security barrier. We stress that our benchmark is for research purposes only, and urge the security community to explore next-generation human verification mechanisms that remain robust against reasoning-driven solvers.

## 6 CONCLUSION

Our work shows that reasoning is a decisive capability for solving modern visual CAPTCHA. With CAPTCHA-X, we pair real-world CAPTCHA challenges with reasoning steps, introduce reasoning-oriented metrics, and propose an agentic pipeline that isolates the role of reasoning. These findings highlight reasoning as central to advancing multimodal AI.

# A APPENDIX

## A.1 DATA COLLECTION

Our data collection approach leverages Selenium (Jason Huggins) and PyAutoGUI (Sweigart) to programmatically interact with websites hosting various CAPTCHA types, including GeeTest challenges (GeeTest) (Gobang, Icon, IconCrush), hCaptcha systems (Intuition Machines, Inc.), VTT (Wang et al., 2018a), and reCAPTCHA V2 (Google). For each CAPTCHA instance, we record comprehensive interaction data during the solving process, capturing all mouse actions with their corresponding screen coordinates. Our annotation scheme covers five distinct mouse action types, including mouse-click, mouse-down, mouse-up, mouse-drag, and mouse-move events. To provide a complete visual context, we capture screenshots both before and after solving each CAPTCHA puzzle, enabling analysis of the initial problem state and solution verification. As shown in Figure 10

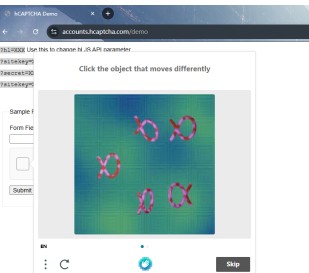

Figure 10: We employ automated tools (Selenium and PyAutoGUI) to collect CAPTCHA images and interaction data during the solving process.

## A.2 REASONING METRICS FOR EACH CAPTCHA TYPE

To complement overall solving accuracy, we further report detailed reasoning-oriented metrics for each CAPTCHA type (Tables 5–10). These include *Reasoning Length* (textual size of generated reasoning), *Reasoning Steps* (discrete step count), *Reasoning Score* (human-annotated quality on a 0–5 scale), *Trajectory Complexity Index* (structural complexity of the predicted action path), and *Reasoning Efficiency* (normalized score relative to reasoning cost). Together, these metrics provide a fine-grained view of how different models trade off reasoning verbosity, structure, and effectiveness across tasks such as Icon, Gobang, and hCaptcha.

Table 5: Reasoning metrics for Icon.

| Model | Reasoning Length | Reasoning Steps | Reasoning Score | Trajectory Complexity Index | Reasoning Efficiency |
|---|---|---|---|---|---|
| Gemini 2.5 Pro | 179.03 | 5.47 | 3.10/5.00 | 0.9697 | 0.843 |
| GPT-O3 | 134.99 | 4.87 | 1.64/5.00 | 0.2052 | 0.387 |
| Claude 4 Opus | 124.74 | 6.21 | 1.43/5.00 | 0.3806 | 0.287 |
| GPT-4O | 88.81 | 6.16 | 1.21/5.00 | 0.3661 | 0.181 |
| GPT-5-Nano | 121.93 | 5.11 | 0.64/5.00 | 0.4022 | 0.000 |
| Gemini-2.0-Flash | 81.16 | 3.34 | 2.32/5.00 | 0.1956 | 1.000 |
| Qwen-2.5VL-72B | 71.93 | 5.13 | 0.75/5.00 | 0.3110 | 0.000 |

Table 6: Reasoning metrics for Gobang.

| Model | Reasoning Length | Reasoning Steps | Reasoning Score | Trajectory Complexity Index | Reasoning Efficiency |
|---|---|---|---|---|---|
| Gemini 2.5 Pro | 287.29 | 8.83 | 3.31/5.00 | 0.9032 | 1.0 |
| GPT-O3 | 110.33 | 6.44 | 1.89/5.00 | 0.8372 | 0.0673 |
| Claude 4 Opus | 104.90 | 7.40 | 2.06/5.00 | 0.6997 | 0.5721 |
| GPT-4O | 118.31 | 6.12 | 1.32/5.00 | 0.1174 | 0 |
| GPT-5-Nano | 90.28 | 4.50 | 1.35/5.00 | 0.1554 | 0 |
| Gemini-2.0-Flash | 148.38 | 2.73 | 1.35/5.00 | 0.4279 | 0 |
| Qwen-2.5VL-72B | 97.96 | 8.70 | 1.28/5.00 | 0.5818 | 0.0588 |

Table 7: Reasoning metrics for hCaptcha.

| Model | Reasoning Length | Reasoning Steps | Reasoning Score | Trajectory Complexity Index | Reasoning Efficiency |
|---|---|---|---|---|---|
| Gemini 2.5 Pro | 276.18 | 8.99 | 2.93/5.00 | 0.9213 | 0.3177 |
| GPT-O3 | 129.69 | 7.36 | 2.50/5.00 | 0.6219 | 0.0359 |
| Claude 4 Opus | 123.43 | 8.67 | 2.52/5.00 | 0.4227 | 0 |
| GPT-4O | 61.53 | 6.37 | 1.93/5.00 | 0.4435 | 0.135 |
| GPT-5-Nano | 119.03 | 4.98 | 2.69/5.00 | 0.3377 | 0.6497 |
| Gemini-2.0-Flash | 69.90 | 5.34 | 2.08/5.00 | 0.5417 | 0.5764 |
| Qwen-2.5VL-72B | 51.89 | 2.69 | 2.20/5.00 | 0.2150 | 1 |

Table 8: Reasoning metrics for GeeTest Space Reasoning.

| Model | Reasoning Length | Reasoning Steps | Reasoning Score | Trajectory Complexity Index | Reasoning Efficiency |
|---|---|---|---|---|---|
| Gemini 2.5 Pro | 171.00 | 6.50 | 3.56/5.00 | 0.9041 | 0.5478 |
| GPT-O3 | 130.20 | 4.55 | 2.00/5.00 | 0.4450 | 0 |
| Claude 4 Opus | 130.55 | 5.13 | 2.71/5.00 | 0.8445 | 0.2393 |
| GPT-4O | 73.00 | 4.00 | 3.24/5.00 | 0.2278 | 0.782 |
| GPT-5-Nano | 63.97 | 1.90 | 2.94/5.00 | 0.2677 | 0.7928 |
| Gemini-2.0-Flash | 62.75 | 3.81 | 3.38/5.00 | 0.0903 | 1 |
| Qwen-2.5VL-72B | 74.87 | 4.62 | 2.55/5.00 | 0.4633 | 0.2901 |

Table 9: Reasoning metrics for Google reCAPTCHA V2.

| Model | Reasoning Length | Reasoning Steps | Reasoning Score | Trajectory Complexity Index | Reasoning Efficiency |
|---|---|---|---|---|---|
| Gemini 2.5 Pro | 215.03 | 8.10 | 3.68/5.00 | 0.8104 | 1.0 |
| GPT-O3 | 104.01 | 7.25 | 2.79/5.00 | 0.3016 | 0.1143 |
| Claude 4 Opus | 153.57 | 8.70 | 2.76/5.00 | 0.6068 | 0.0732 |
| GPT-4O | 89.54 | 6.62 | 2.12/5.00 | 0.3874 | 0.1645 |
| GPT-5-Nano | 145.11 | 7.85 | 2.33/5.00 | 0.3952 | 0.0 |
| Gemini-2.0-Flash | 96.66 | 8.33 | 3.01/5.00 | 0.3909 | 0.8207 |
| Qwen-2.5VL-72B | 45.23 | 6.09 | 2.45/5.00 | 0.1026 | 0.4343 |

Table 10: Reasoning metrics for IconCrush.

| Model | Reasoning Length | Reasoning Steps | Reasoning Score | Trajectory Complexity Index | Reasoning Efficiency |
|---|---|---|---|---|---|
| Gemini 2.5 Pro | 194.40 | 10.71 | 4.04/5.00 | 0.9363 | 1.000 |
| GPT-O3 | 126.01 | 5.54 | 2.81/5.00 | 0.3248 | 0.045 |
| Claude 4 Opus | 135.50 | 10.65 | 3.81/5.00 | 0.5435 | 0.257 |
| GPT-4O | 146.78 | 10.59 | 1.15/5.00 | 0.5258 | 0.053 |
| GPT-5-Nano | 165.54 | 9.00 | 2.94/5.00 | 0.6529 | 0.416 |
| Gemini-2.0-Flash | 115.71 | 9.67 | 2.12/5.00 | 0.5163 | 0.000 |
| Qwen-2.5VL-72B | 120.03 | 11.08 | 1.36/5.00 | 0.4380 | 0.059 |

## A.3 REASONING GENERATION TEMPLATE

We carefully design a **reasoning generation template** that guides the model to generate step-by-step reasoning in a consistent and structured format for our benchmark's reasoning annotations:

**Reasoning Generation Prompt**

Analyze the provided captcha image and the corresponding JSON data which represents the ground truth mouse actions that solve the puzzle.

Your task is to generate a step-by-step reasoning explaining *why* these specific mouse actions (clicks/moves described in the JSON) correctly solve the captcha based on the visual elements and implicit instructions in the image.

Focus on:
1. Identifying the goal of the captcha (e.g., "click shapes that break the pattern", "select all bikes").
2. Explaining how each significant action (usually clicks indicated by mousedown/mouseup pairs) relates to achieving that goal. Refer to the visual characteristics of the clicked elements.
3. Concluding why the sequence of actions is a valid solution.

Captcha Image: [Image provided]

Ground Truth Solution JSON:
```json
{annotation_json_str}
```

Please provide the reasoning STRICTLY as a JSON list of dictionaries. Each dictionary must have an integer "step" key and a string "text" key. Do not include any explanatory text before or after the JSON list itself.
Example Output Format:
[
{{"step": 1, "text": "The puzzle asks the user to click on the 2 shapes that break the pattern."}},
{{"step": 2, "text": "Observing the image, the main pattern consists of dashed-line diamond shapes oriented with points up/down/left/right."}}
... rest of steps
]

Figure 11: Our prompt template.

## B ETHICS STATEMENT

This work adheres to the ICLR Code of Ethics. In this study, no human subjects or animal experimentation was involved. All datasets used, including CAPTCHA-X, were sourced in compliance with relevant usage guidelines, ensuring no violation of privacy. We have taken care to avoid any biases or discriminatory outcomes in our research process. No personally identifiable information was used, and no experiments were conducted that could raise privacy or security concerns. We are committed to maintaining transparency and integrity throughout the research process.

## C REPRODUCIBILITY STATEMENT

We have made every effort to ensure that the results presented in this paper are reproducible. All code and datasets have been made publicly available in an anonymous repository to facilitate replication and verification. The experimental setup, including training steps, model configurations, and hardware details, is described in detail in the paper. We have also provided a full description of our proposed CAPTCHA-X benchmark and evaluation metrics to assist others in reproducing our experiments.

We believe these measures will enable other researchers to reproduce our work and further advance the field.

## D  LLM USAGE

We used Large Language Models (LLMs) exclusively to polish the manuscript's language and readability; all scientific ideas, methodology, and analyses remain the sole responsibility of the authors.

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
