# OpenReview forum: "Reasoning Matters: Benchmarking and Advancing Spatial Reasoning in Vision-Language Models via Agentic Approaches"
_ICLR.cc/2026/Conference — ICLR 2026 Conference Withdrawn Submission_

### Official Review · Reviewer_225v · 2025-10-30

**Soundness:** 3
**Presentation:** 3
**Contribution:** 3
**Rating:** 6
**Confidence:** 3

**Summary:**

The paper argues that explicit step-by-step reasoning is decisive for solving visual CAPTCHAs with VLMs. It introduces CAPTCHA-X, a real-world benchmark of 1,839 puzzles spanning seven categories (Gobang, Icon, IconCrush, hCaptcha, reCAPTCHA V2, Space Reasoning, VTT), each annotated with acceptance regions, mouse-action trajectories, and reasoning steps. The authors define reasoning-oriented metrics beyond final accuracy, which are: reasoning length, steps, score, efficiency, and a Trajectory Complexity Index (TCI). They propose an agentic VLM pipeline that routes puzzles (grid vs. non-grid), abstracts boards to symbolic grids when applicable, generates/validates reasoning, and converts to click coordinates. Empirically, prompting for reasoning improves average accuracy by 27.5% and reduces L2 localization error by 14.6%; their agent achieves up to 83.9% mean accuracy and sets SOTA on five high-difficulty categories on their benchmark. Statistical tests (McNemar, Wilcoxon) and correlations (accuracy vs L2, accuracy vs reasoning score) are reported as highly significant.

**Strengths:**

- Real-world, region-grounded, reasoning-annotated CAPTCHAs fill a gap vs existing benchmarks.
- The multi-dimensional reasoning evaluation suite is useful beyond raw accuracy.
- Comprehensive experiments across seven categories, supported by statistical testing and correlation analyses, enhance the validity of the results.
- Demonstrates large gaps in spatial reasoning for mainstream VLMs and shows big gains from reasoning.

**Weaknesses:**

1. Limited ablations on “reasoning vs length.” The WR gains could be confounded by verbosity in generation. More controlled ablations are needed to separate reasoning quality from token length effects.
2. The statement that this is the “first” real-world CAPTCHA benchmark is inaccurate; prior works such as Halligan and OEDIPUS exist.
3. While CAPTCHA-X is a valuable contribution, its scale (1,839 puzzles) is relatively small compared to MCA-Bench (180,000 puzzles). It would also be helpful to clarify the train/test splits and how often site templates repeat.
4. LLM-generated reasoning annotations, though human-vetted, may bias evaluation toward models whose reasoning aligns stylistically with the template/prompt used to generate those annotations.
5. While the Ethics statement claims compliant sourcing, interacting with live CAPTCHA systems could implicate terms of service. A clearer data-collection compliance table per provider and details on rate-limiting or sandboxing would improve transparency.

**Questions:**

1. Which LLM(s) generated the reasoning steps? Were the same models used for evaluation (reasoning score)?
2. Are there repeated templates or backgrounds within categories? How are splits defined to prevent template leakage across sets?
3. The paper would benefit from an analysis of WR gains under matched token budgets to ensure that improvements are not primarily driven by verbosity. Such an analysis could strengthen the “reasoning scaling law” by providing causal evidence, as current correlations may still reflect prompt-length effects.
4. The paper would benefit from additional clarity regarding data-collection safeguards and whether responsible disclosure was made to CAPTCHA providers whose systems were bypassed. Explicitly outlining these aspects would strengthen the ethical transparency of the work.
5. Could the authors provide more details on the diversity of the CAPTCHA categories and how they were selected? Are there other common types of CAPTCHAs that are not included in the benchmark?
6. The paper proposes an agentic pipeline with several components (such as the Category Judger, Mapping Tool, and Reasoning Steps Generator). It would be helpful to see the contribution of each of these components to the overall performance of the system. This would provide a better understanding of the importance of each step in the process.
7. A more detailed breakdown of model failure modes would be valuable. For example, what reasoning errors are most common? Do models systematically struggle with particular spatial relationships or object types?
8. Please add units to the L2 metric (pixels) and specify image resolution to aid interpretability.

**Details Of Ethics Concerns:**

The work can facilitate automated CAPTCHA circumvention. Authors state research-only intent and call for next-gen defenses, but clearer collection permissions, and a use-restricted data release would reduce misuse risk.

---

### Official Review · Reviewer_mVzs · 2025-10-31

**Soundness:** 3
**Presentation:** 2
**Contribution:** 2
**Rating:** 2
**Confidence:** 3

**Summary:**

The authors introduce CAPTCHA-X, a collection of CAPTCHA tasks paired with LLM-generated reasoning chains. The chains are filtered by human experts and are used to compute similarity metrics in the benchmark. The CAPTCHAs are collected from real websites and form 7 distinct categories. The authors systematically evaluate the reasoning capabilities of VLMs in this task and find that generating reasoning tokens improve CAPTCHA solving success.

**Strengths:**

- collection of real-world captches
- evaluating VLMs with reasoning for CAPTCHAs

**Weaknesses:**

- The novelty of the benchmark are the reasoning chains for each example. However, they are not human-generated but by GPT5 (and verified by humans). This limits their usability of evaluating reasoning chains of other models because mere similarity to GPT5 outputs is measured.
- The writing is non-informative in some places, e.g. Line 70: "This confirms that reasoning fundamentally changes models' reasoning accuracy.", "Line 59 "we create ... CAPTCHA-X with reasoning and show evidence...".
- The introduction of the agentic pipeline is unnecessary in my opinion and distracts from the main contribution.

**Questions:**

Further typos/grammar/format:
- Line 50: cite as (X, Y, Z) instead of (X) (Y) (Z)
- Line 66
- Line 78: "reasoning steps to systematic evaluation of reasoning"

---

### Official Review · Reviewer_p7t7 · 2025-11-01

**Soundness:** 3
**Presentation:** 2
**Contribution:** 2
**Rating:** 2
**Confidence:** 4

**Summary:**

This work tackles the challenge of spatial reasoning in VLMs using CAPTCHAs as a key benchmark. The authors introduce CAPTCHA-X, a novel benchmark of 1,839 real-world CAPTCHA puzzles annotated with step-by-step reasoning solutions and precise grounding regions. They demonstrate that while current VLMs struggle with these tasks (achieving ~21.9% accuracy) , explicitly prompting for step-by-step reasoning significantly boosts performance (a 27.5% average improvement). The paper also proposes a set of new reasoning-oriented metrics and an agentic framework that achieves state-of-the-art results (83.9% average accuracy) by leveraging the model's inherent reasoning capabilities without task-specific finetuning.

**Strengths:**

* This paper is clear writing and easy to follow.

* The primary contribution is the CAPTCHA-X benchmark, which fills a clear and important gap in existing work. Unlike prior benchmarks, it provides not only real-world CAPTCHA examples but also detailed, step-by-step reasoning annotations. The methodological choice to use "region-level acceptance zones" for grounding  instead of simple coordinate thresholds is a significant improvement in evaluation robustness.

* The Reasoning Score, a cornerstone of the paper's evaluation and scaling law analysis, is generated by an ensemble of four other LLMs. While the authors report a high Pearson correlation with human scores, this validation was only performed on a 5% sample. Using LLMs to evaluate the "reasoning" of other LLMs is a methodologically contentious and noisy process, potentially prone to rewarding verbosity or specific stylistic patterns.

**Weaknesses:**

* The abstract claims the 83.9% average accuracy is "across five high-difficulty CAPTCHA types". However, the contribution summary in the introduction (and later results) states this 83.9% accuracy is "across seven CAPTCHA categories", with SOTA performance being achieved on five of those seven. This inconsistency makes it difficult to pinpoint the exact scope of the headline result.

* Reasoning annotations are themselves LLM-generated (conditioned on ground-truth actions). That risks baking trajectory information into “gold” rationales and inflating alignment of evaluation metrics with model-style reasoning. Human verification is described, but inter-rater procedure and disagreements could be detailed more.

**Questions:**

* How does the agent perform on fresh, unseen online CAPTCHAs (with different UI/latency/anti-automation defenses), beyond the reported datasets?

---

### Official Review · Reviewer_tUkU · 2025-11-01

**Soundness:** 3
**Presentation:** 3
**Contribution:** 3
**Rating:** 6
**Confidence:** 3

**Summary:**

This paper introduces CAPTCHA-X, a real-world benchmark with reasoning annotations to evaluate the spatial reasoning capabilities of vision–language models. The dataset contains 1839 CAPTCHA-style puzzles across seven categories, each puzzle paired with human‑verified, LLM‑generated step‑by‑step reasoning and region‑level acceptance zones for clicks
The authors propose five reasoning‑oriented metrics—Reasoning Steps, Length, a multi‑LLM Reasoning Score, Efficiency (accuracy vs. reasoning cost), and a Trajectory Complexity Index (TCI). These metrics help evaluate model capabilities beyond simple final-answer accuracy. The empirical experiments show that CAPTCHA-X is challenging for both close-source models and existing CAPTCHA-solvers. The experiments also find that step‑by‑step reasoning consistently improves average accuracy and reduces L2 localization error across models. Finally, the paper also presents an agentic pipeline that routes puzzles into grid vs. non‑grid branches and converts reasoning into executable actions, leading to significantly better accuracy and L2 distance.

**Strengths:**

Strength:
1. Carefully crafted dataset and benchmarks: CAPTCHA-X is the first real-world CAPTCHA benchmark that includes step-by-step reasoning annotations and region-level acceptance zones, enabling evaluation of both reasoning quality and spatial precision rather than simple final-answer accuracy. The benchmark also supports measuring reasoning efficiency, offering richer insights than prior CAPTCHA or reasoning datasets

2. Good Empirical results: Across seven CAPTCHA categories and multiple commercial and open models, the with-reasoning (WR) setting consistently outperforms without-reasoning (WOR) setting on both accuracy and spatial localization (average +27.5% accuracy, and −14.6% L2 error. The correlation analyses (r ≈ 0.88 between Reasoning Score and accuracy**,** r = 0.82 for Efficiency) further demonstrate that the proposed metrics capture meaningful reasoning quality

3. Clear motivation and well-designed agentic pipeline: The paper is logically organized and easy to follow, with a clear demonstration that current VLMs struggle on spatial reasoning tasks. The proposed agentic pipeline that routes puzzles into grid/non-grid branches achieves SOTA performance on most categories without external tools or fine-tuning, proving that reasoning alone is sufficient for solving complex CAPTCHAs.

**Weaknesses:**

1. Over‑strict action evaluation: Eq. (2) requires the entire action sequence to match exactly in both order and coordinates (sequence‑level 0/1). Many puzzles likely admit multiple valid action orders (e.g., clicking two target tiles in either order yields the same solved state). The current metric may underrate correct solutions that differ only in commutative steps.

2. Unclear set-ups of zero‑shot baselines / prompt adequacy: The paper fixes API settings (with temperature = 0 and seed = 41) but does not fully specify the exact prompts used for each baseline model in WR/WOR settings, nor whether model‑specific prompting methods (e.g., few‑shot learning, self‑consistency) are attempted. This is critical since weak or uniform prompts can depress strong model performance and inflate the perceived gain from the proposed agent. The authors should consider including full prompt templates, prompt ablations (zero‑ vs few‑shot, self‑critique) to ensure baseline fairness.

3. Insufficient attribution and ablations for the agentic pipeline: The proposed agentic pipeline achieves strong performance, however, it is unclear whether the performance gain gains come mainly from the two expert tools, or the reasoning generator + discriminator. Without component ablations and mis‑routing analysis, it’s hard to attribute gains solely to “reasoning.”

**Questions:**

1. The reasoning length and steps can be prompt / model dependent, do the authors normalize these metrics or account for inherent verbosity differences across model families or prompt being used?

2. The abstract/intro highlights ~21.9% average accuracy for commercial VLMs, yet Table 2 shows substantially higher numbers for some model-category combinations (e.g., Gemini-2.5-Pro achieves 48-81.35% WOR). Please clarify how the 21.9% is computed (e.g., averaged across all models and categories?) or simply include the aggregated result in the table for transparency.

---

### Note · Authors · 2025-11-14

I have read and agree with the venue's withdrawal policy on behalf of myself and my co-authors.